# Pre-Entrepreneurs' Perception of the Technology Regime and Their Entrepreneurial Intentions in Korean Service Sectors

## Ilyong Ji [1] and Jinkyung Goo [2,*]

1   Graduate School of IT Convergence Science, Management and Industry,
    Korea University of Technology and Education (KOREATECH), Cheonan 31253, Korea; iyji@koreatech.ac.kr
2   Center for Service Industry, Korea Institute for Industrial Economics & Trade (KIET), Sejong 30147, Korea
*   Correspondence: jkgoo@kiet.re.kr

**Abstract:** Startups and established firms in service sectors mostly fall into the supplier dominated or information intensive categories of Pavitt's taxonomy. Entrepreneurs in these categories are not isolated from the technological environment because they can also be innovative (at least) by adopting technologies from outside. However, it has hardly been studied whether the entrepreneurial intention of pre-entrepreneurs in service sectors can be influenced by how they perceive technological environment. In this paper, using the theory of the planned behavior and technology regime, we examined the role of pre-entrepreneurs' perception of the technology regime (opportunity, accessibility, and cumulativeness) on the formation of entrepreneurial intention in Korean service sectors. The results show that pre-entrepreneurs' perception of the technology regime influences entrepreneurial intention via personal attitude and perceived behavioral control. Opportunity influenced personal attitude and subjective norm; accessibility influenced personal attitude, subjective norm, and perceived behavioral control; and cumulativeness influenced personal attitude and perceived behavioral control. Personal attitude and perceived behavioral control influenced entrepreneurial intention.

**Keywords:** service; entrepreneurial intention; technology regime; theory of planned behavior





## 1. Introduction

In the early twentieth century, Schumpeter [1], in his analysis on the development of a capitalist economy, considered a new combination of factors for the production, credit and capital, and the roles of entrepreneurs as the main factors of economic development. Among these three factors, he put emphasis on the roles of entrepreneurs, and entrepreneurship has emerged as an important field of academic research [2,3]. Now, many countries in the world consider entrepreneurship as a critical source of economic development and make huge investments in support of entrepreneurial activities [4]. In the case of South Korea, the government has been involved in the promotion of entrepreneurship, and the number of startups has increased despite the global pandemic of 2020 [5].

Service sectors are the main playgrounds for entrepreneurs and startup firms. For instance, the total number of startups in South Korea during 2019 and three quarters of 2020 was 2,437,986 according to startup trend statistics of the country [6]. Among the newly established startup firms, about 89% were founded in service sectors, and this trend might not be very different in other countries. Therefore, it is worth examining entrepreneurial intentions in service sectors in order to further promote entrepreneurial activities.

Entrepreneurial intention is one of the popular topics in the field of entrepreneurship research. A large number of studies on this topic are based on Ajzen's Theory of Planned Behavior (TPB) [7], arguing that entrepreneurial intentions are influenced by personal factors such as personal attitude towards entrepreneurship, subjective norms, and perceived behavioral control. While many studies have focused on personal-level factors, including those in TPB, others emphasized that environmental factors were also important. Krueger and Carsrud [8], applying TPB to entrepreneurial intentions, illustrate that exogenous

factors affect entrepreneurial intentions indirectly through personal factors. Some studies have shown that how (pre-)entrepreneurs perceive the external environment plays a role in the formation of entrepreneurial intentions (e.g., [9,10]).

There can be a variety of external environments, such as social, cultural, economic, and technological. Among these, we paid attention to the technological environment in this research. Service sectors are likely to be misunderstood as being isolated from the technological environment. However, firms (and startups) in service sectors mostly fall into the supplier dominated or information intensive categories in Pavitt's taxonomy of technological innovation [11–13], which implies that "they can also be innovative (at lease) by adopting technologies from outside" [14]. For this, it is reasonable to argue that entrepreneurial intentions in service sectors can be influenced by the technological environment.

Technological environments can be expressed in terms of a technology regime. The technology regime is a notion that characterize the technological environment of an industry or a sector, and it can be defined by a specific combination of technological opportunity, appropriability, cumulativeness, and knowledge base [15]. The properties of a technology regime illustrate how the technological environment is favorable for innovation [16], and guides the direction of technological innovation and use [17]. For this reason, a technology regime can influence entrepreneurial intentions in a sector, although indirectly through personal factors in the TPB.

The original notion of the technology regime is that it explains the structural environment for innovation [15]. However, as "different individual entrepreneurs perceive a technology regime differently" [18], different individual perceptions of the external environment may work differently in the framework of TPB. Therefore, we attempt to examine the role of (pre-)entrepreneurs' perception of the technology regime in the formation of entrepreneurial intentions in the service sectors. In the following section, we review the theoretical background and present hypotheses for this research. Then, we will present the research methods and results, and in the last section, we draw conclusions from the results.

## 2. Theoretical Background and Hypotheses

### 2.1. Entrepreneurial Intention

As entrepreneurship and entrepreneurial activities become critical for economic development and sustainability, more scholars have studied the topics of entrepreneurship to date. Among many of the topics, entrepreneurial intention has been one of the most studied. Entrepreneurial intention can be defined as "the commitment to starting a new business" [19].

According to Rai et al. [20], there are three major models of entrepreneurial intentions research, including the entrepreneurial event model, entrepreneurial intentionality model, and TPB model. The first entrepreneurial event model was suggested by Shapero and Sokol [10]. In this model, entrepreneurial intentions are triggered by displacements and are enhanced by desirability and feasibility [10,19]. The second entrepreneurial intentionality model was proposed by Bird [21], and was later further refined by other scholars. According to the model, "contextual (political, social, and economic) factors and personal factors lead to idea/thinking development" [20], which results in entrepreneurial intentionality [20,22]. Later, Boyd and Vozikis [22] refined this model by adding the concept of "self-efficacy". This model is useful as it considers both personal and contextual factors. However, some scholars have pointed out weaknesses in these two models. The entrepreneurial event model focuses on displacements and entrepreneurial events rather than a holistic understanding of entrepreneurial intentions [20,23]. The entrepreneurial intentionality model emphasizes entrepreneurial ideas, but entrepreneurial intention is possible even without ideas [20,23], and the model still requires empirical support.

The third model is TPB. This model was proposed by Ajzen [7]. It was proposed to explain human intention and behavior in general [24], and Krueger and Carsrud [25] applied it to entrepreneurial intention. TPB argues that personal attitude, subjective norms, and perceived behavioral control determine the entrepreneurial intention, and the

intention influences entrepreneurial behavior [7,8,26]. This model has widely been used in entrepreneurial intentions research, and we also adopted it for our research.

Entrepreneurial intention can be defined as the intention "to become self-employed" [24] or "to start a business" [25]. As mentioned above, entrepreneurial intention is influenced by personal attitude, subjective norms, and perceived behavioral control. Personal attitude towards a behavior means "the degree to which an individual has a favorable or unfavorable assessment of the behavior" [7,8,24,26]. In the context of our research, it means "the favorable or unfavorable evaluation of the intention to become an entrepreneur" [27]. The subjective norm is the perceived social pressure from people's opinions about the entrepreneurial behavior [27]. Lastly, perceived behavioral control refers to "propensity to act [27]" or "perceptions of desirability and feasibility underlie career-related decisions" [8]. It is, although controversial, similar to the notion of "self-efficacy" [28,29]. In the context of our research, it can be understood as (pre-)entrepreneurs' "perception on easiness or difficulty of performing the entrepreneurial behavior" [27].

Considering the literature on TPB above, we suggest the following hypotheses:

**Hypothesis 1 (H1).** *Personal attitude toward entrepreneurship positively influences entrepreneurial intention.*

**Hypothesis 2 (H2).** *Subjective norms on entrepreneurship positively influence entrepreneurial intention.*

**Hypothesis 3 (H3).** *Perceived behavioral control positively influences entrepreneurial intention.*

*2.2. The Influence of Perception on Technology Regime*

According to Ajzen [7], and Krueger and Crasrud [8], there are exogenous factors in the TPB model. Exogenous factors are typically either personal or situational variables such as personal traits, demographic, economic climate, and the perceived availability of resources. These exogenous factors either drive attitude, subjective norms, and perceived behavioral control, or moderate the relationship between intention and behavior. Therefore, we can confer that the external environment or an individual's perception of the external environment may work as antecedents of attitude, subjective norms, and perceived behavioral control.

We suggest that an individual (pre-)entrepreneur's perception on the technology regime can be considered as an exogenous factor in the TPB-based entrepreneurial intention model. The notion of a technology regime originates from the works of Nelson and Winter [29,30]. It can be understood as the characteristics of the technological environment "that have major effects on the intensity of innovation, the degree of industrial concentration, and the rate of entry in an industry" [30].

According to Breschi et al. [15], a technology regime can be defined by a set of building blocks. The building blocks are opportunity condition, appropriability condition, cumulativeness condition, and knowledge-base. A particular combination of these building blocks characterizes a technology regime, and hence describes the technological environment [29]. Breschi et al. [15] define the building blocks as described below.

Opportunity condition means "the likelihood of innovation for any given amount of money" [15] invested in innovative activities. A high opportunity condition can facilitate investment or participation in innovation, and under these conditions, new knowledge can be applied to a wide range of products and markets [31].

Appropriability condition is about "the possibilities of protecting innovation from imitation and of reaping profits from innovation" [15]. For instance, appropriability is high if innovations are strictly protected by well-established patent systems, and appropriability is low if innovations and ideas are easily accessible and likely to be imitated. Under high appropriability, established firms can appropriate innovation, while it is hard for new firms or entrants to access innovation. Contrarily, under low appropriability conditions, new firms and entrants can access existing innovations, and therefore this condition may

promote new market entry. For this reason, Malerba [32] used "accessibility", which is the reverse of appropriability. As we performed our research from the perspective of entrepreneurs and startups, we used accessibility instead of appropriability.

Cumulative condition refers to the "extent to which today's innovation depends on past innovation(s)" [15]. In other words, cumulativeness is about whether technological knowledge and capabilities in a field have long been held, improved, and hence accumulated in certain firms or an industry. For instance, the cumulativeness level is high where gradual improvements and incremental innovations based on past innovations are the majority. Contrarily, cumulativeness is low in a technological field that has newly emerged or been characterized by radical innovations.

Lastly, knowledge base refers to the nature of knowledge [15]. Knowledge can be tacit or codified, and generic or specific. Generic knowledge is often associated with basic science, and specific knowledge is associated with applied sciences. The type of knowledge base is also an important factor that characterize the technological environment. However, we did not include knowledge base in our study. We focused on the service sectors, and were especially interested in IT-using ones. In these sectors, the scope of the knowledge base may not be very different, as most of the firms in the sectors are technology (especially IT) users.

Literature on the technology regime argues that the levels and degrees or entrepreneurship may vary across different technology regimes. Winter [33] suggests a distinction between entrepreneurial and routinized regimes. The former is favorable to innovative entry, and the latter is not. Breschi et al. [15], and Malerba and Orsenigo [31] argued that the levels of entrepreneurial entry are different according to the characteristics of the building blocks of the technology regime. For instance, high opportunity conditions, low appropriability conditions, and low cumulativeness facilitate entrepreneurial entry into an industry, but the opposite situations do the opposite. Shane [34] also showed that the dimensions of technology regimes facilitate new firm formation, and Marsili [35] showed that there are distinct combinations of entry barriers and opportunities in routinized and entrepreneurial regimes.

The technology regime is a notion that explains the structural environment for innovation [15], and illustrates the pervasive dynamics in an industry or a sector well [16]. However, we paid attention to individual entrepreneurs' perception of the technology regime. According to Nelson and Winter [30], the technology regime is related to "technicians' beliefs about what is feasible or at least worth attempting". The beliefs are shaped by the nature of technology, and have influences on the decisions for innovation and entry into a field of business. The belief that is pervasive in a sector can be understood as characterizing a technology regime. However, it does not mean that all firms and individuals in a sector perceive a technology regime in a same way. Firms (and individuals) have different levels of "absorptive capacity", which means the firm's ability to recognize the value of new information, assimilate it, and apply it to commercial ends [36]. Because of the presence of absorptive capacity, firms and individuals interpret an environment differently (and hence act differently) [18].

In our research, we suppose that individual entrepreneurs' perception on the three building blocks of the technology regime (opportunity, accessibility, and cumulativeness conditions) act as anteceding factors that drive personal attitude toward entrepreneurship, subjective norms, and perceived behavioral control. We suggest the following hypotheses:

**Hypothesis 4a (H4a).** *Personal attitude toward entrepreneurship is influenced by individual entrepreneurs' perception of opportunity conditions.*

**Hypothesis 4b (H4b).** *Personal attitude toward entrepreneurship is influenced by individual entrepreneurs' perception of accessibility conditions.*

**Hypothesis 4c (H4c).** *Personal attitude toward entrepreneurship is influenced by individual entrepreneurs' perception of cumulativeness conditions.*

**Hypothesis 5a (H5a).** *Subjective norms are influenced by individual entrepreneurs' perception of opportunity conditions.*

**Hypothesis 5b (H5b).** *Subjective norms are influenced by individual entrepreneurs' perception of accessibility conditions.*

**Hypothesis 5c (H5c).** *Subjective norms are influenced by individual entrepreneurs' perception of cumulativeness conditions.*

**Hypothesis 6a (H6a).** *Perceived behavioral control is influenced by individual entrepreneurs' perception of opportunity conditions.*

**Hypothesis 6b (H6b).** *Perceived behavioral control is influenced by individual entrepreneurs' perception of accessibility conditions.*

**Hypothesis 6c (H6c).** *Perceived behavioral control is influenced by individual entrepreneurs' perception of cumulativeness conditions.*

## 3. Methodology

The questions for TPB constructs, including personal attitude, subjective norms, perceived control, and entrepreneurial intention have been developed well in previous literature. We adopted the questions for these constructs from Liñán and Chen [37]. However, in the case of the technology regime, there have been only been a handful of studies that were based on surveys, and it was hard to find a well-established questionnaire. Levine et al. [38] took a survey approach to describe the appropriability of industrial research and development, and Shane [34] modified and used their questions to measure the "effectiveness of patents". Muscio [18], to measure the "perceived technology regime", used two questions for technological opportunity, one for appropriability, and one for cumulativeness. Studies such as Castellacci [39] and Peneder [40] utilized some measures from the Community Innovation Survey of European Union as proxies for opportunity, appropriability, and cumulativeness. In our study, we designed our own questions for opportunity, appropriability, and cumulativeness, considering these and other previous literature.

Opportunity is about "the likelihood of innovation for any given amount of money invested in innovation activity" [15], reflecting the perception of firms and individuals on the ease or potential of innovation [31]. While there is no direct measure for opportunity, Peneder [40] utilized "data on the effort and resources invested in innovation activity". In line with this, Muscio [18] also asked respondents whether investing in a specific direction is convenient and expected to improve products. In our research, we asked whether and how strongly respondents agreed with the following sentences: "Investing in the development or utilization of specific technology may improve my product, service, or business model" and "I have intention to invest in development or utilization of specific technology in order to succeed in my business".

Appropriability is about "the possibilities of protecting innovation from imitation and of reaping profits from innovation" [15]. Breschi et al. [15] measured appropriability using questions about the effectiveness of patents and secrecy to prevent imitation from others. Peneder [40], Muscio [18], and other studies used similar methods of measurement. Here, in our research, we used the term accessibility [32], as we were interested in startups in service sectors where firms are generally technology users, and hence we asked respondents how much they agreed with the following sentences: "It is easy for me to access technologies that are needed for my business" and "If I intend to imitate others' technology, product, service, or business models, I can do this".

Lastly, cumulativeness is about the extent a firm's ability to create new knowledge depends on the existing stock of knowledge [34]. According to Malerba [32], cumulativeness creates first mover advantages and high industrial concentrations. Firms enjoying these advantages introduce incremental innovations based on past or current knowledge [32]. In

other words, high cumulativeness can be characterized by high industrial concentration, incumbents' advantages, and incremental innovation. Hence, we asked respondents how much they agreed with the following sentences: "The development of the technologies that can be considered for my business have been led by a few companies for considerable period of time", "The leading companies are in better positions for the development or implementation of the technologies that can be considered for my business", and "The technologies that can be considered for my business have been developed or improved in incremental ways".

All of the questions were designed to be measured in seven-point Likert scale, ranging from 1 (strongly disagree) to 7 (strongly agree). For data collection, we performed an online web-survey utilizing the service of Macromill Embrain Inc. (http://embrain.com, accessed on 3 July 2021). Macromill Embrain is an online research company in Korea, for which the panel size is over a million across the nation. The company randomly sent emails to panel members, and we collected the responses from the members who answered that they had considered created a start-up in a service sector for the last year. Among a variety of service sectors, we accepted only those utilizing online (or mobile) technologies or other information technologies, and excluded offline-based general services. This was because we wanted to see the influence of the technology regime more clearly. To improve respondents' understanding, we provided examples of startups utilizing online (or mobile) technologies or other information technologies. For instance, examples such as online markets for second-hand products, online or mobile education, mobile-based delivery-food restaurants, IT-based business services, and travel information services were given.

The survey was conducted from 30 July to 3 August 2020, and we collected a total of 219 responses. For the data analysis, we used SPSS AMOS 22.0 software. The variables of our analysis were opportunity (OPP), accessibility (ACC), cumulativeness (CUM), personal attitude (ATT), subjective norm (SN), perceived behavioral control (PBC), and entrepreneurial intention (EI).

## 4. Results and Discussion

### 4.1. Results of Analysis

The respondents' profiles are summarized in Table 1. Among the 219 respondents, 104 (47.5%) were male and 115 (52.5%) were female. In terms of age, 31.5% of them were in their 20s, 32.9% were in their 30s, 22.4% were in their 40s, 10.5% were in their 50s, and 2.7% were in their 60s or above. While respondents were distributed over all adult age groups, there were more young respondents, as those in their 20s and 30s represented over 60% of respondents. This may reflect the current trend in Korea, which emphasizes young entrepreneurship. The majority of the respondents held graduate degree (74.0%), and about 10.0% held post graduate degrees. About 36.5% were employees, and there were also students (10.5%), self-employed (10.0%), and others. Among the 219 people, 168 had no prior experience in startups, and the rest had one or more prior experience.

Before the hypothesis tests, we tested the reliability of the items. As the results in Table 2 show, Cronbach's α for most variables (OPP, CUM, PA, SN, PBC, and EI) were over 0.8. One variable (ACC) showed a Cronbach's α of 0.791, which is just slightly below 0.8. These results show that the constructs secured a high internal reliability.

To test the validity of the measurements, we performed a confirmatory factor analysis using AMOS. Based on the results, we tested the convergent validity and discriminant validity. For the convergent validity, standardized λ over 0.5, average variance extracted (AVE) over 0.5, and C.R value over 0.7 were suggested in general. As shown in Table 3, the standardized λ value of all of the constructs exceeded 0.5, and C.R. values were above the 0.7 level. The AVE values for constructs, except for PA and PBC, were above 0.5. AVE values for PA and PBC were slightly below 0.5, but could be accepted because the C.R values for the constructs were far above 0.6 [41,42]. For the discriminant validity, the AVE value for a construct should be greater than the square of correlation with other constructs. Table 4 shows that the result met this criterion.

**Table 1.** Descriptive Statistics.

| Category | | Frequency | % |
|---|---|---|---|
| Gender | Male | 104 | 47.5 |
| | Female | 115 | 52.5 |
| Age | 20s | 69 | 31.5 |
| | 30s | 72 | 32.9 |
| | 40s | 49 | 22.4 |
| | 50s | 23 | 10.5 |
| | 60s or above | 6 | 2.7 |
| Education | High school | 35 | 16.0 |
| | Graduate degree | 162 | 74.0 |
| | Post graduate degree | 22 | 10.0 |
| Occupation | Student | 23 | 10.5 |
| | Employee | 80 | 36.5 |
| | Professional employee | 19 | 8.7 |
| | Self-employed | 22 | 10.0 |
| | Professional self-employed | 11 | 5.0 |
| | Housekeeping | 15 | 6.8 |
| | Other | 49 | 22.4 |
| Previous Startup Experience | 0 | 168 | 76.7 |
| | 1 | 40 | 18.3 |
| | 2 | 7 | 3.2 |
| | 3 or more | 4 | 1.8 |

**Table 2.** Reliability analysis results.

| Construct | N.Q. | Cronbach's $\alpha$ |
|---|---|---|
| OPP | 2 | 0.898 |
| ACC | 2 | 0.791 |
| CUM | 3 | 0.824 |
| PA | 5 | 0.840 |
| SN | 3 | 0.834 |
| PBC | 6 | 0.912 |
| EI | 6 | 0.918 |

**Table 3.** Convergent validity.

| | | | Estimate | S.E. | C.R. | Standardized Estimate | AVE | C.R. |
|---|---|---|---|---|---|---|---|---|
| PA_1 | ← | PA | 1 | | | 0.632 | | |
| PA_2 | ← | PA | 1.295 | 0.164 | 7.875 | 0.627 | | |
| PA_3 | ← | PA | 1.077 | 0.121 | 8.897 | 0.733 | 0.451 | 0.802 |
| PA_4 | ← | PA | 1.325 | 0.135 | 9.787 | 0.842 | | |
| PA_5 | ← | PA | 1.33 | 0.141 | 9.452 | 0.798 | | |
| SN_3 | ← | SN | 1 | | | 0.812 | | |
| SN_2 | ← | SN | 1.03 | 0.086 | 11.918 | 0.833 | 0.529 | 0.770 |
| SN_1 | ← | SN | 1.091 | 0.099 | 10.986 | 0.745 | | |

**Table 3.** *Cont.*

| | | | Estimate | S.E. | C.R. | Standardized Estimate | AVE | C.R. |
|---|---|---|---|---|---|---|---|---|
| PBC_6 | ← | PBC | 1 | | | 0.803 | | |
| PBC_5 | ← | PBC | 1.158 | 0.076 | 15.319 | 0.881 | | |
| PBC_4 | ← | PBC | 1.177 | 0.078 | 15.168 | 0.875 | 0.495 | 0.853 |
| PBC_3 | ← | PBC | 1.126 | 0.076 | 14.878 | 0.863 | | |
| PBC_2 | ← | PBC | 1.033 | 0.081 | 12.774 | 0.773 | | |
| PBC_1 | ← | PBC | 0.832 | 0.091 | 9.151 | 0.591 | | |
| OPP_2 | ← | OPP | 1 | | | 0.891 | 0.792 | 0.884 |
| OPP_1 | ← | OPP | 1.001 | 0.065 | 15.329 | 0.915 | | |
| ACC_2 | ← | ACC | 1 | | | 0.771 | 0.596 | 0.746 |
| ACC_1 | ← | ACC | 1.08 | 0.101 | 10.681 | 0.849 | | |
| CUM_3 | ← | CUM | 1 | | | 0.724 | | |
| CUM_2 | ← | CUM | 1.189 | 0.11 | 10.775 | 0.838 | 0.550 | 0.785 |
| CUM_1 | ← | CUM | 1.203 | 0.116 | 10.383 | 0.784 | | |
| EI_6 | ← | EI | 1 | | | 0.915 | | |
| EI_5 | ← | EI | 0.934 | 0.043 | 21.796 | 0.908 | | |
| EI_4 | ← | EI | 0.932 | 0.044 | 20.943 | 0.894 | 0.525 | 0.867 |
| EI_3 | ← | EI | 0.758 | 0.057 | 13.356 | 0.717 | | |
| EI_2 | ← | EI | 0.864 | 0.065 | 13.322 | 0.716 | | |
| EI_1 | ← | EI | 0.718 | 0.058 | 12.369 | 0.684 | | |

**Table 4.** Discriminant validity test.

| | Correlation | | | | | | AVE |
|---|---|---|---|---|---|---|---|
| | PA | SN | PBC | OPP | ACC | CUM | |
| PA | 1 | | | | | | 0.451 |
| SN | 0.527 (0.278) ** | 1 | | | | | 0.529 |
| PBC | 0.412 (0.170) ** | 0.359 (0.129) ** | 1 | | | | 0.495 |
| OPP | 0.552 (0.305) ** | 0.351 (0.123) ** | 0.433 (0.187) ** | 1 | | | 0.792 |
| ACC | 0.488 (0.238) ** | 0.338 (0.114) ** | 0.626 (0.392) ** | 0.65 (0.243) ** | 1 | | 0.596 |
| CUM | 0.495 (0.245) ** | 0.203 (0.041) ** | 0.453 (0.205) ** | 0.57 (0.325) ** | 0.558 (0.311) ** | 1 | 0.550 |
| EI | 0.647 (0.419) ** | 0.374 (0.140) ** | 0.616 (0.379) ** | 0.532 (0.283) ** | 0.552 (0.305) ** | 0.517 (0.267) ** | 0.525 |

( ) = square of correlation, ** $p < 0.05$.

Considering the results above, we can conclude that the measurement model of our research achieved sufficient reliability and validity.

We conducted a structural equation model (SEM) analysis using AMOS 22.0. The results of the model fitness analysis indicated that $\chi^2$ was 643.939(df = 309) and normed $\chi^2$ ($\chi^2$ divided by the degree of freedom) was 2.084, which was close to 2. In addition, GFI (=0.814), CFI (=0.915), IFI (=0.915), and TLI (=0.903) were all close to or above 0.9. These results suggest a good model fitness. The results of the SEM analysis are displayed in Figure 1.

The results of the hypothesis test are displayed in Table 5. Hypotheses 1 to 3 are about the influence of the three constructs of TPB on EI. PA and PBC appeared to be antecedents of EI, and therefore hypotheses 1 and 3 were supported. However, the influence of SN on EI failed to meet statistical significance as it had a p value of 0.872. Hypotheses H4a, H4b and H4c are about the influence of technology regime on PA. OPP, ACC, and CUM all showed a significant influence on PA, and therefore H4a, H4b and H4c were all supported. Hypotheses H5a, H5b and H5c are about the impact of technology regime on SN. OPP and ACC had a significant influence on SN (supporting H5a and H5b), but the influence of CUM on SN was not significant at a 0.05 level. Lastly, hypotheses H6a, H6b and H6c are about the technology regime and PBC. The influences of ACC and CUM on PBC were statistically significant, but OPP's influence was not at the 0.05 level. Therefore, H6a was rejected.

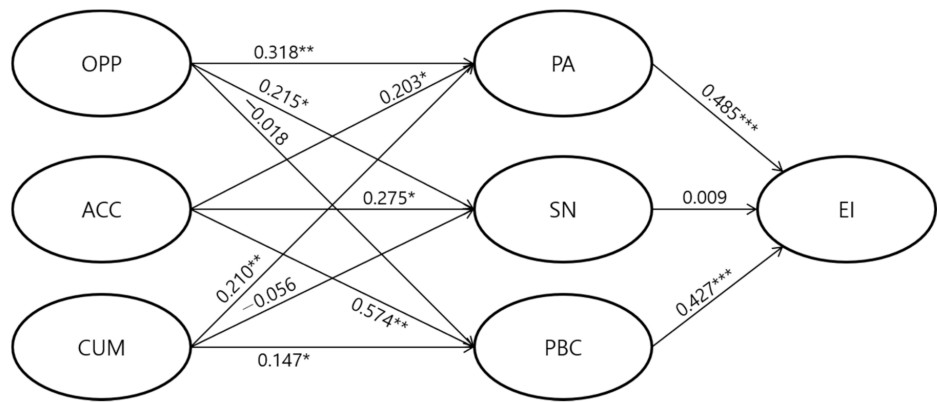

**Figure 1.** SEM results. Values are standardized estimates, *** $p < 0.000$, ** $p < 0.05$, and * $p < 0.1$, (model fit) $\chi^2 = 643.939$ (df = 309), $p = 0.000$, normed $\chi^2/\mathrm{df} = 2.084$, GFI = 0.814, CFI = 0.915, IFI = 0.915, and TLI = 0.903.

**Table 5.** Hypotheses tests.

|  |  | Estimate | Standardized Estimate | S.E. | C.R. | *p* | Results |
|---|---|---|---|---|---|---|---|
| H1 | PA→EI | 0.62 | 0.485 | 0.104 | 5.972 | *** | supported |
| H2 | SN→EI | 0.008 | 0.009 | 0.048 | 0.161 | 0.872 | rejected |
| H3 | PBC→EI | 0.361 | 0.427 | 0.058 | 6.244 | *** | supported |
| H4a | OPP→PA | 0.221 | 0.318 | 0.071 | 3.124 | 0.002 | supported |
| H4b | ACC→PA | 0.154 | 0.203 | 0.08 | 1.923 | 0.055 | supported |
| H4c | CUM→PA | 0.178 | 0.21 | 0.079 | 2.257 | 0.024 | supported |
| H5a | OPP→SN | 0.227 | 0.215 | 0.119 | 1.903 | 0.057 | supported |
| H5b | ACC→SN | 0.316 | 0.275 | 0.14 | 2.257 | 0.024 | supported |
| H5c | CUM→SN | −0.072 | −0.056 | 0.134 | −0.539 | 0.59 | rejected |
| H6a | OPP→PBC | −0.019 | −0.018 | 0.102 | −0.187 | 0.852 | rejected |
| H6b | ACC→PBC | 0.656 | 0.574 | 0.13 | 5.025 | *** | supported |
| H6c | CUM→PBC | 0.189 | 0.147 | 0.114 | 1.655 | 0.098 | supported |

*** $p < 0.000$.

### 4.2. Discussion

According to the results for hypotheses 1 to 3, subjective norm (SN) did not have an influence on entrepreneurial intention (EI), despite the established theory (TPB). However, it is not unusual to find previous studies that report weak or insignificant influences of the construct on entrepreneurial intention. For instance, Autio et al. [43], Liñán [44], and Fayolle et al. [45] found only a weak or insignificant influence of subjective norms on entrepreneurial intention. Moreover, a number of studies that were performed in Korea reported similar results. Lee [46], in her study of entrepreneurial intention for agricultural business in Korea, found that subjective norms were not a significant factor for entrepreneurial intention. In studies by Han and Cho [47], and Yue et al. [48], the same hypothesis (subjective norms have a positive relation with entrepreneurial intention) was rejected. Until recently, we had not found a common explanation for the result. Some scholars attempt to find a reason from TPB itself [43,49], and others supposed or implied that cultural differences may be one of the reasons [48,50]. We suggest that this issue should be further studied in the future.

Regarding technology regime's role in TPB, all hypotheses except for hypothesis 2 were accepted. This means that a pre-entrepreneur's perception on the technology regime may exert some influences on personal attitude, subjective norms, and perceived behavioral control. This roughly supports the argument by Krueger and Crasrud [8], that exogenous factors affect entrepreneurial intentions "indirectly". However, not all

elements of the technology regime affect all the personal attitudinal factors of TPB (personal attitude, subjective norms, and perceived behavioral control). According to the results, the influence of opportunity (OPP) on perceived behavioral control (PBC) and the influence of cumulativeness (CUM) on subjective norm (SN) were not statistically significant. Why did opportunity exert an influence on personal attitude (PA) and subjective norms, but not on perceived behavioral control? In addition, why did cumulativeness exert an influence on personal attitude and perceived behavioral control, but not on subjective norms?

We suppose that the meanings of the constructs provide some hints for the questions. Regarding the first question, the meaning of perceived behavioral control may be the key. Ajzen [7,28] developed TPB from the theory of reasoned action. What makes TPB different from the theory of reasoned behavior is that TPB has the construct of perceived behavioral control. According to Krueger and Carsrud [8], perceived behavioral control reflects an individual's perceived feasibility of performing a behavior or personal situational competence (for instance, self-efficacy), while the other two constructs (personal attitude and subjective norm) reflect the perceived desirability of performing a behavior. As opportunity condition of the technology regime implies the exploration of the room, niche, or potential for innovation, a high opportunity may increase personal attitude and subjective norms, but may not exert a direct influence on perceived behavioral control. Rather, it can be supposed that whether an individual has confidence in his/her access to resources or capabilities may enhance perceived behavioral control.

Regarding the second question, we suppose that the meaning of cumulativeness is important. Cumulativeness is related to whether technological knowledge and capabilities in a field have been long held, improved, and hence cumulated in certain firms or an industry. High cumulativeness can be characterized by high industrial concentration and incremental innovation [32].

Under a high cumulativeness condition, technological environment is stable, and it is quite clear what technology is available and who has the technology. From the perspective of technology users in service sectors, it is easier to identify where to get technological support for implementing their business under a high cumulativeness condition (accessing, using, or imitating the technology may be a matter of accessibility or appropriability rather than that of cumulativeness). Therefore, high cumulativeness may improve the personal attitude of technology-using pre-entrepreneurs in service sectors, and increase their perceived behavioral control. However, subjective norms are related to other people's evaluation of the entrepreneurial behavior of an individual. We suppose that simply knowing what to use and where it is may not be effective for improving other people's evaluation (while knowing what the opportunity is and how to access it may be so).

## 5. Conclusions

The purpose of this research was to examine the role of pre-entrepreneurs' perception of the technology regime in the formation of entrepreneurial intention in Korean service sectors. More specifically, we attempted to examine the influence of pre-entrepreneurs' perception on the three building blocks of technology regime (opportunity, accessibility, and cumulativeness) on personal attitude, subjective norms, and perceived behavioral control in TPB. For this purpose, we adopted questions for TPB constructs from previous literature, and designed a set of questions about the perception of the technology regime.

The results show that pre-entrepreneurs' perception on the technology regime influences entrepreneurial intention via personal attitudinal factors. Opportunity influenced personal attitude and subjective norms, accessibility influenced all three personal attitudinal factors (personal attitude, subjective norms, and perceived behavioral control), and cumulativeness influenced personal attitude and perceived behavioral control. There were no significant influences of opportunity on perceived behavioral control and of cumulativeness on subjective norms. Personal attitude and perceived behavioral control influenced entrepreneurial intention, but subjective norms did not show a significant influence on entrepreneurial intention. Regarding subjective norms, it is not unusual that the influence

of the construct on entrepreneurial intention is only weak or insignificant in previous literature. We suggest that there should be further studies about this issue.

This result implies that the technological environment can have an exogenous influence on entrepreneurial intention, even in the service sectors. Firms and startups in the service sectors can be misunderstood as being isolated from the technological environment, but they are under the influence of the technological environment because they are at least users of technologies, as suggested by literature on Pavitt's taxonomy [11–14]. Pre-entrepreneurs' perceptions of technological opportunity, accessibility, and cumulativeness may affect entrepreneurial intention in service sectors, although indirectly through personal factors. Therefore, it is important for policy makers and practitioners supporting startups to improve pre-entrepreneurs' perception of the technology regime. For instance, providing information about what technologies are available and how they can be used for service business, where and in what state the technologies are, and how the technologies can be accessed can be effective at improving the perceptions.

This study is meaningful, as it is one of the few attempts to integrate entrepreneurial intention studies with innovation studies. We introduced the notion of the technology regime (which is one of core theories of innovation studies) in the model of TPB. Moreover, we paid attention to individual entrepreneurs' perception of the technology regime. To our knowledge, there has been only one paper [18] before our research that dealt with perception of the technology regime, and most literature regarded the technology regime as one of the structural factors. In addition, this study provides a more detailed view about the influences of the technology regime on entrepreneurial intention by examining the roles of the three building blocks of the technology regime. Lastly, while previous studies paid little attention to technology regimes in the service sectors, we concentrated our attention on the service sectors.

Despite the value above, this study also has some limitations. First, the scope of our analysis was limited to Korean service sectors. In particular, we focused our interest on IT-using services. There should be further studies on other sectors in the future. Second, the number of samples was only 219, which was just enough for our analysis. Studies with a larger sample size are required. Third, the relationship among constructs must be further examined in the future.

**Author Contributions:** Conceptualization, I.J.; methodology, J.G.; formal analysis, J.G.; investigation, I.J. and J.G.; resources, I.J. and J.G.; writing—original draft preparation, I.J.; writing—review and editing, I.J. and J.G.; visualization, J.G.; supervision, J.G.; project administration, I.J.; funding acquisition, I.J. All authors have read and agreed to the published version of the manuscript.

**Funding:** This work was supported by the Ministry of Education of the Republic of Korea and the National Research Foundation of Korea (NRF-2019S1A5C2A02082342).

**Institutional Review Board Statement:** Not applicable.

**Informed Consent Statement:** Not applicable.

**Data Availability Statement:** The data presented in this study are available on request from the corresponding author.

**Conflicts of Interest:** The authors declare no conflict of interest.

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
