# Peer review of "Pre-Entrepreneurs’ Perception of the Technology Regime and Their Entrepreneurial Intentions in Korean Service Sectors"

_2199-8531, doi:10.3390/joitmc7030179_

Round 1

Reviewer 1 Report

This paper is very interesting and showed pre entrepreneurs' perception on technology regime influences entrepreneur intension via personal attitude and perceived behavioral control.  

This paper discussed hypotheses 1-3, SN does not have an influence on EI in the case of Korea, then, it is because of cultural differences compared from literature reviews.  More explanation and analysis on the issue of cultural differences on the results will be necessary for better understanding. 

Author Response

Dear Reviewer

We appreciate your review and comment on our manuscript. Reflecting your opinion, we revised some parts of the manuscript.

Your comment was:

This paper discussed hypotheses 1-3, SN does not have an influence on EI in the case of Korea, then, it is because of cultural differences compared from literature reviews. More explanation and analysis on the issue of cultural differences on the results will be necessary for better understanding.

Our reply:

In the previous version, we thought the issue was related to Korean cultural context. However, while revising the manuscript to reflect your comment, we found that the issue (insignificant influence of subjective norm on entrepreneurial intention) had appeared not only in Korea but also in other countries. In addition, some scholars are still working on this issue. Therefore, we decided that we should not conclude too hastily. We re-wrote a part of discussion section (lines 357-369, p.10). We provided more cases of the issue and explanations, and made a suggestion that there must be further studies on this issue. Accordingly we revised a part of conclustion section (lines 426-429, p.11), and added 6 more references in the last page (lines 546-562).

Thank you again for your valuable comments.

Reviewer 2 Report

The main content of research presented in the paper is an analysis of the role of pre-entrepreneurs’ perception on technology regime in the formation of entrepreneurial intention in Korean service sectors.

The topic is not unique, but it is worthy of researching.

The main proposed approach are questions for TPB constructs from previous literature, and designed a set of questions for perception of technology regime.

The deduced conclusions based on the research methods show that pre-entrepreneurs’ perception on technology regime influences entrepreneurial intention via personal attitudinal factors.

The conclusions are tenable. However, in my opinion, it is not clear what progress has been made compared with the current research results. I advise the authors to focus on this point better.

The abstract is informative. It reflects the body of the paper.

The introduction provides sufficient background information for readers in the immediate field to understand the problem.

The text is well arranged and the logic is clear. There is virtually no grammatical errors in the article. The related concepts are introduced clearly. The readability is sufficient.

The proposed approaches are not new. The novelty lies in its application to a concrete situation.

The theoretical analysis in this article is sufficient.

The figure and all tables are clear enough to summarize the results for presentation to the readers. The figure and all tables are well referred to in the text.

However, the caption for Table 3 appears below the table rather than above it, as in all other tables.

The reference section is informative. In the references section not all references are carefully formatted. References to websites are very incomplete.

The references section has reference 49. empty. Authors should delete it.

Author Response

Dear Reviewer

We appreciate your review and comment on our manuscript. Reflecting your opinion, we revised some parts of the manuscript.

Your comments were:

(1) However, in my opinion, it is not clear what progress has been made compared with the current research results. I advise the authors to focus on this point better.

(2) However, the caption for Table 3 appears below the table rather than above it, as in all other tables.

Our reply:

To reflect your comment(1), we re-wrote a part of conclusion section. In lines 443-452 (p.12), we tried to show what the values of our research are.

To reflect your comment(2), we moved the caption above the table.

Thank you again for your valuable comments. The comments were very helpful for us to improve our paper.